

# A New Perspective and Explanation to the Formation of
# Plasmaspheric Shoulder Structure
Hua Zhang [1]    Guangshai Peng[1]    Chao Shen [2]
[1]Institute of Space Weather, Nanjing University of Information Science & Technology,
Nanjing, China.
[2]Harbin Institute of Technology, Shen Zhen, China.
*Correspondence to:* Hua Zhang (289534957@qq.com)
**Abstract**
Over the hours of 5-9 UT on June 8 2001, the extreme ultraviolet (EUV) instrument
onboard IMAGE satellite observed a Shoulder-like formation in the morning sector
and a Plume-like structure straddling in the between noon and dusk region.
Simulation results of the plasmapause formation based on mechanism of drift motion
called Test Particle Model (TPM) and have reproduced various plasmapause
structures and subsequent evolution of the Shoulder. The analysis indicated that the
Shoulder is created by a dawn-dusk convection electric field intensity, sharp reduction
and spatial nonuniform manifested. As, combination of the plasmaspheric rotation rate
speed up with L-shell increase and plasma flux do radial outflow in the predawn
sector to interact, and produce an asymmetric bulge that rotates eastward. The
Shoulder-like structure rotates sunward and develops to the single or double Plume
structure during active times.
Keywords: plasmapause; shoulder-like; plume-like;IMAGE/EUV
## 1. Introduction
The plasmasphere is important region in the inner magnetosphere, surrounding the
Earth and extending to 5 Earth radii(Re), which contains dense($10$-$10000$ cm$^{-3}$) and
cold plasma (below 1ev). The plasmapause formed by a superposition of corotation
and convection electric field in the inner magnetosphere (Nishida,1966; Chen and
Wolf, 1972). The formation and size of plasmapause varies with geomagnetic activity
level. Generally, as the disturbance level increasing, the plasmapause position closer



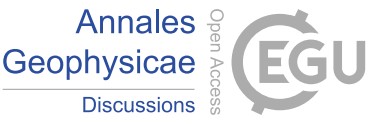

to the Earth and of shape deviate from circle in the equatorial plane (Grebowsky,
1970). Atypical plasmapause structures, such as 'bulge' and Plume occur often in both
whistler and in-situ data (Carpenter and Anderson,1992). There are many theoretical
researches study to explanation of the formation of Plume (Grebowsky,1970; Pierrard
and Lemaire, 2004; Zhang et al., 2013).
The EUV instrument onboard IMAGE satellite has launched in March, 2000,
which provided a global perspective to the plasmasphere, such as Plume, Finger,
Notch and Shoulder, and so on, some of plasmaspheric structures observed by EUV
(Sandel et al., 2001). One of plasmaspheric structures, Shoulder, has less study in the
previous papers than Plume. But, the Shoulder may play important role on a loss
mechanism for the ring current (Burch et al., 2001). So, it is important to study the
formation mechanism of Shoulder.
At present, there are no convincing explanations for dynamic formation of
Shoulder. Goldstein et al.(2002) firstly proposed an explanation, based on the
Magnetospheric Specification Model(MCM) simulation output, for the formation of
the Shoulder. They presented that the Shoulder is created by sudden decrease of
dusk-dawn electric field. As interplanetary magnetic field (IMF) turns northward
from southward, trigger antisunward flow of plasma in predawn sector, to produce
an asymmetric bulge called Shoulder. Later, based on physical mechanism of
interchange instability and a Kp-dependent E5D electric field model, Pierrard and
Lemaire (2004) suggested that the Shoulder is not the result of radial outflow of
plasma, same as the presentation of Goldstein et al. (2002) , but is inward plasma
drift in post-midnight sector.
Then, scarce papers about dynamical formation of the Shoulder are delivered than
of Plume. In this paper, we used TPM to simulate dynamical formation of the
Shoulder, using Weimer's statistical E-field (Weimer, 2001; Zhang et al., 2012),
which is both spatially nonuniform and dynamically responsive to change
geomagnetic and solar wind conditions. To drive the TPM model, several inputs are
used: Dst; solar wind (SW) and interplanetary magnetic field (IMF) data sets. The
authors make attempt to a new convincing explanation for formation of the





Shoulder-like structure, different from the previous explanations.
**2. Shoulder Observation**

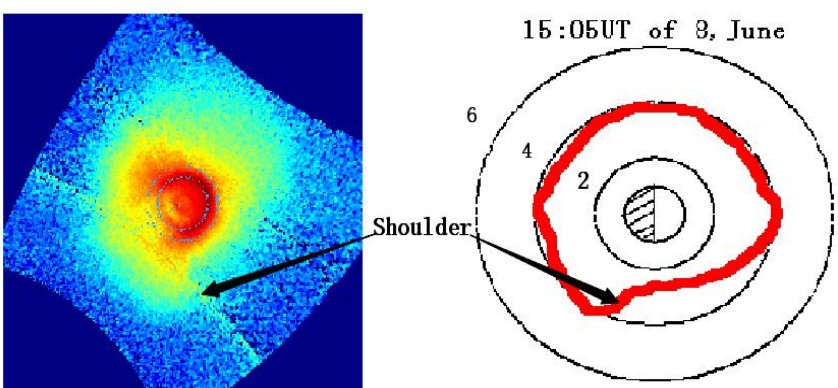


**Figure1. Snapshot of plasmasphere(left panel) by EUV instrument, at 15:05 UT of 8 June 2001,**
**Sunlight is incident from the upper right. Earth is in the center of panels and Shoulder is**
**observed and labeled in the snapshot. Right panel is plasmapause of that extracted from left**
**plasmapheric image**.
The Figure 1 illustrates the Shoulder-like structure, a sharp radial plasmaspheric
structure about 1 RE radial extension, in the post-midnight sector, which was viewed
by EUV imager onboard IMAGE satellite at 15.05 UT of 8 June 2001. The right panel
illustrates the plasmapause extracted from the left panel in the Figure1, and the outer
boundary of plasmasphere is assumed to be 40% of maximum brightness of 30.4nm
$He^+$ emission, where the intensity is the logarithm of the luminosity (Pierrard and
Cabrera, 2006). Then, the Shoulder-like is labeled and marked by arrows in the plot.
Subsequent pictures show that the Shoulder-like structure remaining and corotating
with main plasmaspheric body by discussion in the next section. That is mean the
outer edge of the Shoulder corotates faster than the inner edge in development phase
(Goldstein et al., 2002). Then, the Shoulder moves eastward to afternoon sector and
evolves into the Plume-like structure. Over the next hours, the outer body of Plume
flows sunward from noon sector, resulting in the Plume thinned out and disappeared
(can see simulation of Figure 3). In the next section, we would discuss simulation of
Shoulder and Plume evolution on 8 June 2001 case base on the TPM method.



## 3. Simulation

In region of plasmasphere occupied, charged particles are cold plasma (e.g. energy of particles is several eV or less). So, we can assume that plasma elements have only $\boldsymbol{E} \times \boldsymbol{B}/B^2$ drift motions (Li and Xu, 2005; Lejosne and Mozer, 2016). Here, the electric field intensity of E-model is superposition of convection and corotation electric field. Electric field plays a key role on plasma drift motion and the formation of plasmasphere (Pierrard et al., 2008). In the present paper, we use the Weimer's convection electric field (Weimer, 2001) to model the magnetospheric convection electric field (Zhang et al., 2012), and T96 magnetic field to model the background magnetic field.

In the simulation, the calculation regions are radial range of 2-7 Re and azimuthal span 0-359°. Dispersion by iso-spacing grids that correspond to the radial and azimuthal steps are equal to 0.1Re and 1° respectively, in the magnetic equatorial plane. Ten particles are placed into each grid, so particle density is proportional to $L^{-1}$ which is not consistent with the actual density in a saturation state (close to true density presumably is proportional to $L^{-3}$), but is adequate to study the evolution of plasmaspheric morphology using a skeleton map of particles during a substorm period.



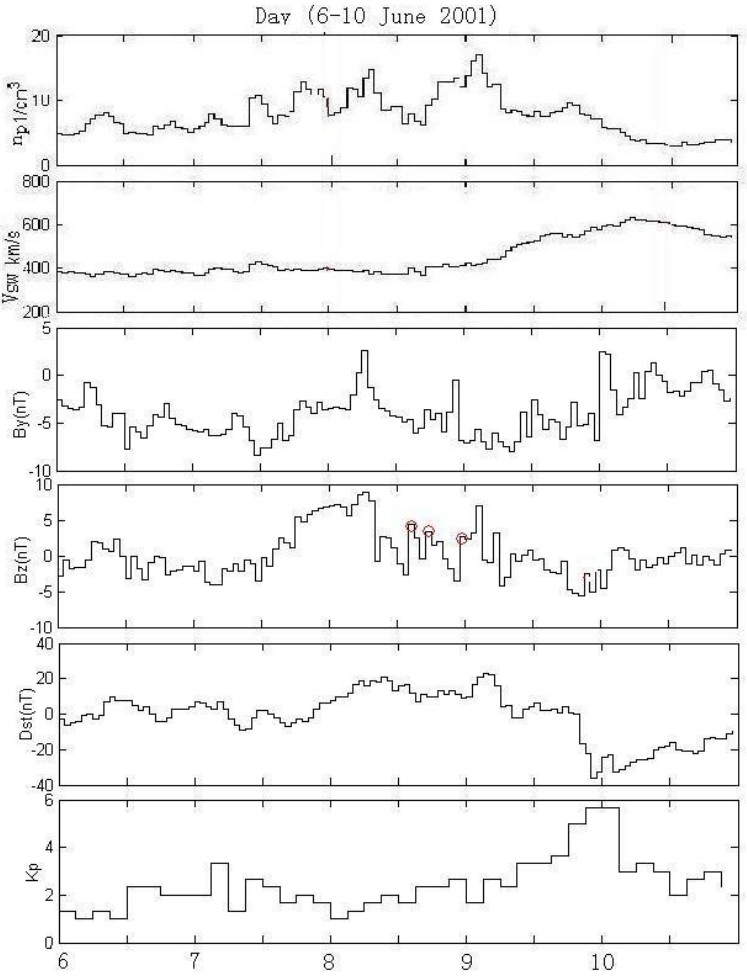


**Figure 2. Input parameters of TPM model, the variation of the By and Bz component of the IMF,**

**the Dst index and Kp index, on 6 -10 June 2001, is a typical substorm case.**



The paper presents the case of 8-9 June 2001, to study the evolution of the
shoulder and propose a hypothetical explanation produced by TPM simulation.
During the geomagnetic substorm, all of the TPM inputs are available. IMF and Solar
Wind data are available in ACE satellite data center, and Dst index can see in Word
Data center for Geomagnetism, Kyoto. Fig.2 shows the By, Bz components of the
IMF, the Dst index and the geomagnetic activity index Kp, observed over 6 to 10 June
2001. This is a typical substorm case that Kp index gradually increases up to 5+ and
then decreases. The TPM run with 3-minute time resolution from 6 June at 00:00 UT



to 10 June at 12:00 UT. The results of simulation are showed in Fig.3, which
corresponding times are labeled on the title of each panel. Comparison of TPM
simulation (black body) and EUV observation (red line) in Fig.3, the simulated
plasmapause positions correspond generally rather favorable with the EUV
observations. The results show that the plasmapause is seldom smooth or irregular,
due to the fluctuations in plasmapause region cause by successive particles injection
during a disturbance period (Goldstein et al., 2002; Gallagher et al., 2005), verified by
simulation and EUV observation, in agreement with previous whistler observations
(Carpenter and Anderson,1992). In addition, observations and simulations are not
identical, due to deviation in the extraction of the boundary from EUV image and
optical contamination of the image (Sandel et al., 2001; Zhang et al., 2013).



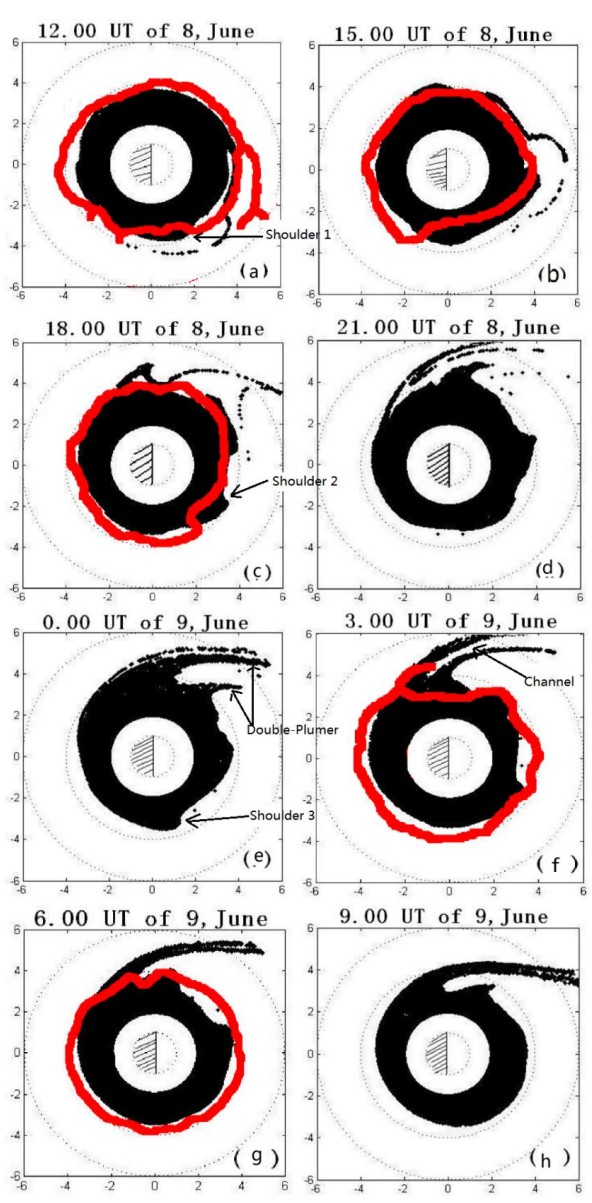


**Figure 3. The simulation of plasmaspheric morphology compare with EUV/IMAGE observation in the**

**geomagnetic equatorial plane on 8 - 9 June 2001. The red irregular curves indicate the plasmapause**

**observation by EUV. The dotted circles on the panels correspond to L=1, 2, 4 and 6.**

Panels of Fig.3(a) - (h) illustrate the plasmasphere obtained on the interval of from
8 June at 12:00 UT to 9 June at 09:00 UT 2001, and every three hours output a



snapshot. The results of the simulation show that the evolution and development of
the features of the plasmapause, like Shoulders and Plumes. One can see that the
plasmapause is sharper and becomes closer to the Earth in the predawn sector. The
reason is the increase of rotation velocity resulting in plasmapause of peeled off in the
predawn sector (Pierrard and Cabrera, 2006; Verbanac et al., 2018). At 15.05 UT of 8
June, the TMP simulation captures a infant Shoulder-like structure in panel Fig.3 (b),
and then corotates with the plasmasphere body moved eastward and further
reproduces a mature Shoulder formation in Fig.3(c). The overall agreement between
TPM simulation and EUV observed is quite well, but the TPM Shoulder is located
~1.5 hours earlier in magnetic local time (MLT) that probably originated from the
convection electric field model (Goldstein et al., 2002; Pierrard and Cabrera, 2005;
Zhang et al., 2013).

139       The EUV observation illustrated in Fig.3 (f) shows that a Plume is indeed observed

in the afternoon or dusk sector. The results of the simulation also reproduce the
formation and the motion of the Plumes,which derive from the Shoulder structure,
illustrated in panels of Fig.3 (d)-(f). The simulation show that the Shoulders generate
in the post-midnight sector (Verbanac et al., 2018), and then rotates eastward around
the Earth to the afternoon sector (Goldstein et al., 2002). When the level of
geomagnetic activity increase, the plasma element in the Shoulder around the outer
plasmasphere would convection outward and then into the dayside magnetopause (Li
and Xu, 2005; Pierrard et al., 2008), and produce the plasmaspheric Plume structure.
The Shoulder1 firstly arises at 12 UT in the morning sector( see in Fig.3(a)), and then
corotates with the Earth reaching to the afternoon region at 18 UT ( see in Fig.3(c)),
on 8 June 2001. At this time, Kp index increases to 3+ ( see in Fig.2), and
magnetosphere convection slightly enhance that trigger plasma elements in the
Shoulder1 doing sunward convection, then produce the Plume1 at 21 UT on 8 June
2001 (see in Fig.3(d)). The mature Shoulder2, illustrated in Fig.3(b), corotates
eastward with the Earth to the afternoon-dusk sector. During period of 0-3 UT on June
9, Kp index gradually increases up to 5+, indicating that magnetospheric convection
is enhanced and the convective electric field increases. The infantile Plume2,



illustrated in the panel of Fig.3(e), derives from outflow of plasma elements in the
Shoulder2, and evolves into the mature Plume2 in Fig.3(f). Later, the double-plumes
formation that is extension from the plasmapause to the magnetosphere, presented in
the simulation results in panels of Figs.3 (e)-(f).

161        The cavity in between the double Plumes, or between Plumes and the main body

of plasmasphere, may be responsible for the formation of Channel and Notch
structures (Gallagher et al., 2005). The base and the westward edge of the Plume is
connected with the main body of plasmasphere. And there is a cavity topology, a
low-density region, between the tail structure of the plasmasphere and the main body
of plasmasphere. That is the channel structure of the plasmasphere. The Plume
corotates with the Earth to become thinner, and disappear finally (Li and Xu, 2005).
The plasma refilling from plasma sheet results in the Notch structure disappear
(Gallagher et al., 2005). The results of simulation show the Channel structure in
Fig.3(e)-(f). Gallagher et al. (2005) proposes that Notches and Channels share same
origin, which derive from a low-density cavity in the dusk region during recovery at
the base of the plasmaspheric Plume. The absence of Notch structure in this
simulation event, due to the fact that the potential structure does not cause the inward
convection of plasma in the afternoon sector, and the low disturbance time is
maintained for no long enough time.

176        By contrastive analysis on between Fig.2 and Fig.3, the formation of the

Shoulder is produced during the intensity of the convection electric field sudden
decrease (Goldstein et al., 2002; Pierrard and Lemaire, 2004), when IMF turns
northward. There are three Shoulders produced during this substorm period, depicted
in panels of Fig.3 (b)-(g). The time of the Shoulder appearance are labeled by three
red circles in Fig.2, at 14:00 UT, 17:00 UT, 23:00 UT on 8 June respectively. At
moment, the Bz component of the IMF turns northward. But, not all of the times as
the Bz component of the IMF turns northward, could produce the Shoulder structure.
The Bz value must lower than previous 24-hours value, due to the intensity of the
convection electric field lower than previous level, so the last closed equipotential line
(LCE) would close to the Earth and result in plasmapause of peeled off in the



predawn sector (Zhang et al., 2013). One can see that no shoulder appearance in the
results of the simulation, produced at 02:00 UT, 05:00 UT, and 08:00 UT on 9 June
2001 respectively.
4. **Discussion**
The physical explanation of Shoulder formation is not yet understood. In present
section, we use the case of Fig.1 as an example to investigate the physical mechanism
of Shoulder formation based on the TPM model. Fourteen test particles are placed in
the range of $2.5 \leq L \leq 3.8$, initial position locate at 12:00 MLT, space step takes 0.1Re,
and then trace these particles motion. Outputs are the trajectory (see in Fig.4(a)) and
the rotation rate (see in Fig.4(b)) of these test particles corresponding to given
magnetic local time and universal time illustrated in the bottom of Fig.4.

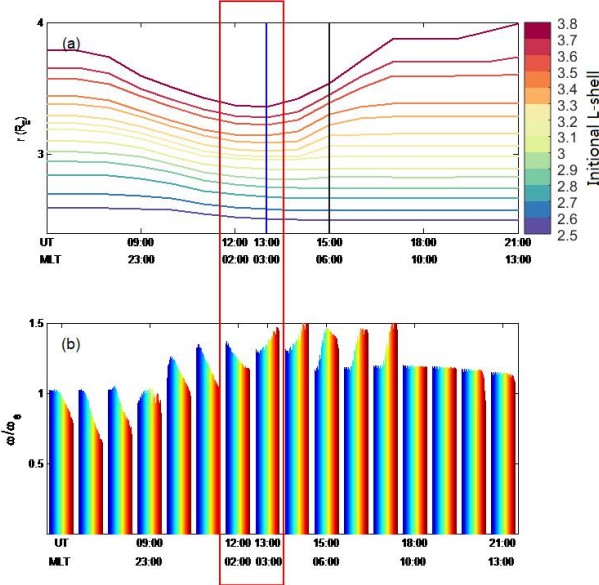


**Figure 4. The trajectory (upper plot) and the rotation rate (bottom plot) of 14 test particles**
**corresponding to given magnetic local time and universal time during a substorm, to explain the**
**physical mechanism of Shoulder formation. Fourteen test particles are placed in the range of**
**$2.5 \leq L \leq 3.8$, initial position locate at 12:00 MLT, and space step takes 0.1Re**



The top panel shows that the outer part of plasmasphere (L>3.3 Re) drift inward in
the before 02:00MLT sector, and move outward (could reach up to 3.9 Re position) in
the predawn sector (after 03:00MLT sector) (Verbanac et al., 2018). The radial motion
of inner plasmasphere (L<3.3) is negligible. So, the Shoulder has a sharp eastern edge
about 0.5Re~0.7Re in radial extension and in a range of 3 MLT. Goldstein et al.(2002)
proposed the shoulder formation by an outward radial motion of plasma in a narrow
range and in the morning sector. The conclusions of Goldstein (2002) and Verbanac
(2018) verify the simulation of this paper.
The lower panel shows the corotational angular velocity of test particles in the
range of 2.5 < L< 4.0. The simulation results suggest that plasma element in
plasmasphere region rotation speed varies significantly with radial distance (Galvan,
2010). The inner part of plasmasphere rotates faster than its outer part in before 02:00
MLT sector, vice versa in a range of in the 03:00-08:00 MLT sector [Lejosne and
Mozer, 2016]. The previous researchers analyze the EUV observation and propose the
Shoulders structure have MLT sharpening in the angular direction, which indicate the
outer edge of the Shoulder rotates faster than the inner edge, resulting in the gradual
increase of MLT-profile of the Shoulder (Goldstein et al., 2002). The lower panel
shows, with the increase of L, the rotation rate of the plasmasphere tends to slightly
decrease on the dusk side and obviously increase on the dawn side.
Fig. 4 indicate, in the region of 21:00 - 23:00:00 MLT, that the rotation rate is
about corotation in the inner plasmasphere (L<3), but is the interval of 70% - 90% of
corotation in the outer plasmasphere (L>3). The rotational value decreases with the
increase of L [Galvan et al., 2010]. Gallagher et al. (2005) investigates the drift rate of
notches in the geomagnetic quite phase, and the results show that the average rotation
rate of plasmasphere is about 90% of the corotational rate, in agreement with the
results of Lejosne and Mozer (2016). When the plasma elements rotate to the region
of 23:00 - 02:00 MLT, rotation rate in the outer plasmasphere reaches to ~ 130% of
corotation, and in the inner plasmasphere is also close to the corotation rate. The
results show that the rotation rate of plasmasphere is overall increasing in the region.
When the plasma elements rotate into the region of between 03:00 and 08:00 MLT,



the plasma elements in the outer plasmasphere move outward and have a radial
outflow of about the interval of 0.2 - 0.7 Re. In addition, the plasma elements in the
outer plasmasphere rotate faster than the inner plasmasphere in this region. The
Fig.4(b) shows that rotation rate in the outer plasmasphere highly reaches to ~ 140%
of corotation, and rotation rate in the inner plasmasphere is close to 110% of
corotation. So, we suggest that the physical mechanism of shoulder formation is the
result of plasma extrusion in the predawn sector, caused by outer plasmasphere drifts
radial outward and rotates faster. In present paper, the results show that the rotation
rates of simulation are higher than the observations, and not consistence with Huang
et al. (2011) and Galvan et al. (2010). The first reason is that the level of Kp index and
the convection of magnetosphere is increase, so the value of these parameters driven
convection field in this case is greater than the previous study articles in the
geomagnetic quite case (Galvan et al., 2010; Huang et al., 2011 ; Verbanac et al.,
2018 ). And the second reason is that the model does not include the shielded electric
field, which results in a larger total electric field value in calculation (Goldstein et al.,
2002; Pierrard et al., 2008).

250       The dawn-dusk asymmetry of convective electric field is caused by the terminal

conductivity gradient of the ionosphere. The subrotation of the ionosphere drives the
subrotation of the plasmasphere, and the plasmaspheric drift is correlated with the
phase of geomagnetic storm (Burch et al., 2004). The convection electric field of
Weimer (2001) is obvious dawn-dusk asymmetry, that causes a smaller increase on
the downside and a lager decrease on the duskside, indicating that the subrotational
effect of the plasmasphere is modulated by field-aligned current changes and
conductance variations (Liemohn et al., 2004). The asymmetry of potential pattern
causes the sunward convection in the magnetospheric night-side to be larger than that
in the morning side, resulting in the subcorotational flow in the dark side and the
supercorototional flow in the morning side (Gallagher et al., 2005).

**5. Conclusion**



In this paper, we have simulated the case of substorm on 8 June 2001 to investigate
the physical mechanism of Shoulder formation based on TPM model that utilizes
Weimer's electric field and the drift motion theory. We use the E-model and the
B-model are qusi-static background field and global averages. So, the results of
simulation have some deviations with EUV observation. But, we have satisfactorily
reproduced the evolution and development of the features of the plasmapause, like
Shoulders and Plumes. And then, the physical mechanism of Shoulder formation has
been investigated. The following results are obtained:
1.The formation of Shoulder is association with IMF northward turning in the
predawn sector. But not all of IMF northward turning could produce shoulders. It
is only that Bz of IMF value must lower than previous 24 hours value.
2.The physical mechanism of Shoulder formation is the result of plasma extrusion
in the predawn sector, caused by outer plasmasphere drifts radial outward and
rotates faster.
3.The formation and evolution of Plumes have also been study in this paper. One
can see single or double Plumes appear in the dusk or afternoon sector, and then
become thinner with time, finally disappear. A second-Plume derives from the
Shoulder which rotates to the afternoon sector and convects into the outer
magnetosphere and then forms the second-Plume.
At this model, we not consider the refilling process of ionosphere. In the future
work, the refilling process should be considered, expect to obtain more perfect results
comparing with EUV observations. And also, the physical mechanisms of
plasmaspheric features observed by EUV/IMAGE, like Notch or Channel, also are to
investigate by TPM model in future work underway.
**Author contributions:** Zhang H. conceptualized the project and wrote the original
draft of the paper. Peng G. S. modified the Figures and coded Fortran program. Shen C.
supervised the project, and reviewed and edited the paper.

**Acknowledgment**: The author thanks the professor D. R. Weimer, who provided the
code of Weimer's electric field model and ACE satellite data center and Word Data



center for Geomagnetism, Kyoto provided observation data. The dataset of
EUV/IMAGE could download from website http://euv.lpl.arizona.edu/euv.

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
