# Peer review of "A New Perspective and Explanation for the Formation of"

_Annales Geophysicae, 2020_

## Referee Comment (RC1) · Anonymous Referee #1 · 1 Jan 2021

General comments:

The authors proposed a new theory to explain the formation of the plasmaspheric shoulder structure. The authors presented an event study using a combination of observations from the EUV/IMAGE and Test Particle Model (TPM). The authors suggested that the shoulder was created by (1) "a dawn-dusk convection electric field intensity, sharp reduction and spatial nonuniform manifested," and (2) a different plasmaspheric corotation rate. The theory to explain the formation of the plasmaspheric shoulder is interesting. However, the English in the present manuscript is not of publication quality and requires significant improvement. The evidence provided in the current manuscript is not sufficient to support the main conclusions. It is recommended that the authors carefully proofread the manuscript and provide further evidence to support

their conclusions. Please see the detailed comments below.

Specific comments:

1. The manuscript is poorly written, and the expressions in many sentences are confusing. These mistakes made the manuscript hard to understand. However, it is highly recommended that the authors carefully proofread the manuscript.

2. Figure 3 illustrates the comparison between the observations and the TPM model, which is essential to the main conclusions. However, the authors provided only the processed plasmapause location (red curves) every 3 hours. It is recommended that the authors (1) show the raw images from the EUV/IMAGE observations for comparison, (2) show the simulations at higher temporal resolution (e.g., 1 hour) so that the evolutions are clear.

3. The authors discussed the formation of the double Plumes in the TPM model. However, they did not provide any observations to validate the existence of the double Plume.

4. The proposed theory of the plasmaspheric shoulder involved the dawn-dusk convection electric field. It is recommended that the authors provide the comparisons between the Weimer electric field and the EUV/IMAGE observations and the TPM model, which is essential to support the conclusion.

5. Captions for Figures 2 and 3 need further improvement. The red circles in Figure 2 are barely visible. The definition of the black/white filled contours in Figure 3 are missing. Some legends are missing from Figure 3 (e.g., Plume2 in line 158).

6. Line 191-197 and Figure 4 are very confusing. Are these test particles placed in a static electric field at a specific time (the same as Figure 1)? Or are the electric field changing during the substorm event (from 0600 UT to 2100 UT)? Is the x-axis time-dependent (UT) or location-dependent (MLT)?

7. Figure 4b is very confusing and hard to understand. I suggest that the authors

consider a contour plot (w/w) with the x-axis (either UT or MLT) versus the y-axis (L shell).

8. Line 277-281 (conclusion 3): The third point is more of a result from the TPM model rather than a scientific conclusion. The authors should provide (1) a scientific intensive in the introduction section, (2) provide observational evidence to support the formation and evolution of the Plume (or double Plume, or second-Plume), and (3) show a comparison between the observations and the simulation to support their conclusion.

9. Line 119-120. The reasons also include the limitation in the TPM model and the unrealistic Weimer electric field model.

Technical corrections: 1. Confusing sentences or grammatical errors:

1) 'a', 'an', 'the' are missing throughout the manuscript.

2) The sentence in lines 16-18.

3) The sentence in lines 73-74

4) The sentence in lines 79-80

5) Line 105: Word->World

6) The sentence in lines 79-80

7) Line 109: run->runs

8) Line 110: which-> whose

9) The sentence in lines 148-150

10) Line 156: the infantile Plume2. What does 'infantile' mean?

11) The sentence in lines 168-169

12) The sentence in lines 148-150
13) The sentence in lines 175

14) The sentence in lines 184-187

15) The sentence in lines 208-210

16) The sentence in lines 218-220

17) The sentence in lines 239-240

18) The sentence in lines 244-246

19) Line 255: downside->dawnside

20) The sentence in lines 218-220

---

## Referee Comment (RC2) · Anonymous Referee #2 · 16 Jan 2021

The manuscript describes an alternate method for reproducing plasmaspheric shoulders that have been modeled previously. Based on their Test Particle Model (TPM), it is suggested that a new mechanism is observed to form shoulders. Language usage needs to be significantly improved in the abstract and body of the manuscript. The conclusions, insofar as I understand the text, are not adequately supported. As a consequence the manuscript does not demonstrate a significant contribution to the field. Figure 4 does not appear to contribute to the conclusions derived from it. Additional detail about the TPM simulation are needed, in addition to discussion of simulation limitations and artifacts resulting from its design. Contrary to the conclusions, use of the Weimer empirical electric field model in TPM appears to preclude making conclusions about the physical processes that cause subcorotation of the plasmasphere.

[Figure]

Specific concerns are described in the comments. I do not recommend publication of the manuscript in its present form.

1. Line 83: Plasmasphere ions are defined in the introduction as having energies of less than 1 eV. Here the definition given is that the plasmasphere consists of "several eV or less". The two descriptions need to agree; <1 eV is generally used, but there is flexibility. Citing a source for whatever is used is worthwhile.

2. Line 85: It is stated that the intensity of the electric field model is a superposition of the convection and corotation electric fields. The electric field model this line refers to is not directly stated. Line 54 states that the TPM uses the Weimer statistical electric field model. The Weimer model is empirical, based on observations. It is not a superposition of simple electric fields. The extent to which the Weimer model accurately represents measured electric fields and those measurements accurately represent actual electric fields, then this empirical model incorporates all the physical processes that produce large to small-scale features in inner magnetospheric electric fields. It also means that nothing about the underlying physical processes are available to be determined by use of the Weimer model in the TPM simulation. This point is of particular importance in the Section 4 Discussion and in the conclusions stated for the paper.

3. Line 94: Only 10 particles per simulation box is quite course. What are the boundary conditions for the simulation? Are particles allowed to leave or enter the simulation to maintain the number within the simulation? Only black and white is used to represent model results in Figure 3. If black means there is at least one particle in a simulation box, then that needs to be stated.

4. Line 96: Why is it stated that the density variation goes as $L^{-3}$? Most authors report a variation of approximately $L^{-4}$. Whatever is used here needs to be justified in the text or by citation.

5. Lines 114-116: Can this simulation produce a smooth plasmapause boundary when there are so few particles in the simulation? What is considered to be smooth given

the small number of particles in each simulation box?

6. Lines 114-116: What particle injections are referred to here? An "injection" of particles would normally be expected to come from outside the simulation, whether along the field or transverse to it. Quantitatively, what does smooth or irregular mean as it is used here and how can it be "seldom smooth or irregular" as stated?

7. Line 129: How is a sharper plasmapause boundary model result shown in Figure 3? "Sharper" has previously been used to qualitatively refer to the density gradient across the plasmapause. The black and white representation of the model result shown in Figure 3 cannot show a gradient. Small irregularities in the plasmapause can be seen in Figure 3, however this may be due to the small number of particles in simulation cells, a modeling artifact rather than a physical result.

8. Lines 129-131: The model result in Figure 3 does not show peeling off of plasmaspheric plasma in the predawn region. The formation of a shoulder does not constitute a peeling off of the plasmasphere. The plumes evident in all panels of Figure 3 show the plume extending sunward in afternoon and early evening magnetic local time. I am unaware of any observation of the outer plasmasphere being peeled away at predawn magnetic local times. The post-dawn outer plasmasphere has been found to sometimes contribute to a broad early-plume that then narrows to afternoon MLT only.

9. Lines 352-354: I do not find this reference in the journal as cited. Could there be an error in the citation?

10. Lines 141-142: The plume features shown in Figure 3 exist before the shoulder convects into afternoon magnetic local times where the plume(s) is(are) connected to the plasmasphere. The shoulder is first indicated in Figure 3c near 9 hours magnetic local time. In panel (d) three hours later a plume is forming between roughly 16 hours and 18 hours magnetic local time. This shoulder feature has not azimuthally convected further than about 13 hours MLT. Another 3 hours later in panel (c) a plume appears to be forming where the shoulder has come to be located. That does not mean the

shoulder had a causal role in forming the plume. It is more likely it only happened to be there when geomagnetic activity increased, which changed the global convection pattern in your electric field model so as to form a new plume that would have formed whether the shoulder was there or not. A specific explanation must be provided in order to substantiate the statement that the shoulder is functionally responsible for the plume as currently stated.

11. Lines 142-143: The simulation shows that the TPM simulation for the conditions during this event period resulted in a shoulder forming in at post-midnight MLT. One simulation cannot establish a pattern of shoulders forming at post-midnight MLTs as currently stated.

12. Lines 148-150: The feature at 12 hours MLT in Figure 3a appears to be a remnant plume originally formed in at afternoon MLT due to earlier activity. It does not have the characteristics of a shoulder. The discussion in the last paragraph on page 8 is at least poorly expressed if not also poorly conceived, as suggested in the previous few comments. It needs to be corrected or removed.

13. Line 168: Notch structures and the outer plasmasphere do not refill from the plasma sheet as currently stated in the text. The injection of plasma ions discussed by Gallagher et al. (2005) refers to a potential source of meso-scale electric field due to charge separation in the injected energetic plasma. It is suggested in that study that this meso-scale electric field may cause the interior "W" shaped feature.

14. Line 173: What is meant by "inward convection?" Convection inward to lower L-shell does not appear to happen during storm-time recovery. Isopotential contours are not axially symmetric, however, therefore there can be inward and outward radial motion of the plasmapause without a change in plasmasphere content. The dusk bulge is an example.

15. The first paragraph of the Discussion section: Figure 4a shows paths taken by semi-corotating plasma, but does not show the formation of a sharp radial change

in the plasmapause as stated. Goldstein et al. (2002) state that their model shows the shoulder forming across a narrow 3-5 hour MLT region. Figure 4a shows a gradual outward motion of a radial group of particles from 3-9 hours MLT. For a shoulder to form there must be a transient and narrow MLT region where plasma is differentially moved in L-shell, which cannot be shown with the test particle simulation presented. Even if an adjacent and leading parcel of plasma did not move outward in L-shell after 1300UT, only a difference of 0.1-0.2L appears to have taken place between 3-5 MLT, not 0.5-0.7L as stated. Figure 3b-c show shoulder-2 formation much better, though a shorter time interval between these panels would do that much better. In fact a sequence of model images between these two might provide a more useful display than the current Figure 4a. Reversal of relative semi-corotation with L-shell shown in 4b is interesting as an explanation for steepening the leading eastward edge of a shoulder.

16. Line 220: "increase of the MLT-profile of the shoulder" does not say what is needed here. Perhaps "steepening of the MLT-profile of the shoulder" would be a better word to use.

17. Lines 247-260: Which model does not include electric field shielding in the inner magnetosphere? Is this referring to your TPM? If so, the statement is not substantiated in the text. While shielding is not explicitly included in the Weimer electric field model, the fact that the Weimer model is empirical means that the model includes whatever physical processes are active. That will include shielding if it is happening, as discussed in previous papers. The Weimer model, hence TPM, provide no information about the physical processes taking place that produce the measured electric field.

18. Lines 259-260: Gallagher et al. (2005) specifically report not finding a day-night asymmetry in subcorotational flow. They also do not report finding supercorotational flow, only speculate that asymmetry in the dawn-dusk convection pattern may cause net subcorotational motion.

19. Conclusions: I posit that you cannot investigate the physical mechanisms for shoulder formation using the Weimer empirical electric field model, as that model only represents the measured net electric fields resulting from whatever physical processes are involved in their formation without distinction for those processes. Please substantiate how this can be done.

20. Conclusion 1: It has not been demonstrated that IMF Bz must be lower than the previous 24 hours for a shoulder to form. A statistical study or theory is required before there is adequate basis for the conclusion. The statement on lines 184-186 is an observation that might be used to suggest correlation or dependence, but no more than that.

21. Conclusion 2: The conclusion does not add to what has previously been found. It is incumbent on you to be clear how this new work contributes in the context of previous work. This statement does not do that.

22. Conclusion 3: No significance has been established between the position of a shoulder and the formation of a plume connected to the plasmasphere at that location. Given that plumes form at the onset of convection enhancement, which is not connected to the earlier formation of a shoulder, the presence of a shoulder where a plume begins to form is likely no more than coincidence. It is well established that plumes form in the afternoon/dusk region without the presence of a shoulder feature.

23. Line 75: "Shoulder-like structure" is acceptable, but "shoulder-like" is not used by itself and if it is shoulder-like, then it would be better to simply refer to it as a shoulder. Lines 66-68 refer to the shoulder structure and define it in words and in Figure 1. That is adequate to subsequently refer to it as a "shoulder". Unless you consider the feature you are referring to as something different from what has previously been described as a plasmaspheric shoulder, then I recommend you simply use that description or just shoulder.

24. Lines 12-14: This sentence is not grammatically correct. Perhaps the authors intend it to be something like, "The plasmapause formation is simulated using the

Test Particle Model (TPM), which is based on drift motion, which reproduces various plasmapause structures and evolution of the Shoulder feature."

25. Lines 14-18: These sentences are grammatically incorrect. English language usage needs to be improved throughout in the paper. No further comment about that will be made in this review.

26. "Plume" is misspelled in Figure 3e as "plumer".

---

## Author Comment (AC1) · 9 Mar 2021

Dear anonymous Referee1:

I am very happy to receive your recommendation and very grateful for your advice. We have followed your comments to revise this manuscript. Then, due to the stupid organization and poor English make readers understand difficulty, we have made efforts to revise and hope that you could be satisfied. In the resubmitted paper, new text is emphasis as red text.The Referee Comments is abbreviated to "RC", and Authors' Response is abbreviated to "AR".

The following are the response of each major comment: RC 1: The manuscript is poorly written, and the expressions in many sentences are confusing. These mistakes

made the manuscript hard to understand. However, it is highly recommended that the authors carefully proofread the manuscript. AR 1: I am agree with the advice, and have revised this problem in my manuscript. We will call for professional company to polish the manuscript before formal publication.

RC 2: Figure 3 illustrates the comparison between the observations and the TPM model, which is essential to the main conclusions. However, the authors provided only the processed plasmapause location (red curves) every 3 hours. It is recommended that the authors (1) show the raw images from the EUV/IMAGE observations for comparison, (2) show the simulations at higher temporal resolution (e.g., 1 hour) so that the evolutions are clear. AR 2: To recommendation (1): the raw images of the EUV/IMAGE observations are color drawing and have serious light contaminates (see in left panel of Figure 1), so no processing to superpose the simulations of TPM is not good effects. We submit the raw images of the EUV/IMAGE observations in the supplementary material. To recommendation (2): In this case, there are 8 panels output in Figure 3. If 1 hour temporal resolution is used to simulate, there are 24 panels outputs in Figure 3 results in crowds and poor typesetting. During two adjacent panels of the TPM output, the plasmasphere corotates approximately 3MLT, and spatial resolution enough is used to study evolution of plasmaspheric structures.

RC 3: The authors discussed the formation of the double Plumes in the TPM model. However, they did not provide any observations to validate the existence of the double Plume. AR 3: The double Plumes firstly arises in Figure 3(e), but the IMAGE satellite is too close to the Earth to provide any global view of the plasmasphere during this period of time. The double Plumes structure has been simulated in Pierrard and Cabrera [2006] ( has been listed in References). In this paper, the author indicates the double plumes derives from the Shoulder evolution based on sequential panels of TPM simulation.

RC 4: The proposed theory of the plasmaspheric shoulder involved the dawn-dusk convection electric field. It is recommended that the authors provide the comparisons between the Weimer electric field and the EUV/IMAGE observations and the TPM model, which is essential to support the conclusion. AR 4: I am sorry that I cannot understand the referee's meaning. The Weimer electric field maps into magnetosphere as the dawn-dusk convection electric field, and then is used to simulate evolution of the plasmasphere in the TPM model.

RC 5: Captions for Figures 2 and 3 need further improvement. The red circles in Figure 2 are barely visible. The definition of the black/white filled contours in Figure 3 are missing. Some legends are missing from Figure 3 (e.g., Plume2 in line 158). AR 5: Thank you for your recommendation, I have revised Figures 2 and 3 according to your advice in the resubmitted manuscript. And the definition of the black/white filled contours in caption of Figure 3 rewrite.

RC 6: Line 191-197 and Figure 4 are very confusing. Are these test particles placed in a static electric field at a specific time (the same as Figure 1)? Or are the electric field changing during the substorm event (from 0600 UT to 2100 UT)? Is the x-axis time-dependent (UT) or location-dependent (MLT)? AR 6: I am sorry for indistinct presentation in Line 191-197 to confuse the referee. These test particles placed in a static electric field and the electric field changing with 3-minute time resolution (same as describe in line 109-110 ). The x-axis is both time -dependent( UT) and location-dependent (MLT). I have rewrite the caption of Figure 4 .

RC 7: Figure 4b is very confusing and hard to understand. I suggest that the authors consider a contour plot (w/w) with the x-axis (either UT or MLT) versus the y-axis (L-shell). AC 7: I have rewrite the caption of Figure 4. The legend illustrates various initial location of test particles. The Formation of Shoulder derived from the rotation of differential motion withÂăL-shell, so the y-axis label as rotation rate is necessary .

RC 8: Line 277-281 (conclusion 3): The third point is more of a result from the TPM model rather than a scientific conclusion. The authors should provide (1) a scientific intensive in the introduction section, (2) provide observational evidence to support

the formation and evolution of the Plume (or double Plume, or second-Plume), and (3) show a comparison between the observations and the simulation to support their conclusion. AC 8: I am agree with the advice, and have revised this problem in my manuscript. I have introduced Pierrard and Cabrera (2006) to the introduction in line 33-34, who also simulated the double Plumes in paper, but not explained origin of second-Plume. I also revised Figure 3 (f) to produce the observations and the simulation of the double Plume.

RC 9: Line 119-120. The reasons also include the limitation in the TPM model and the unrealistic Weimer electric field model. AC 9: I revised this problem of manuscript according to your advice. Please see revised content in Line 122-123.

Technical corrections: Confusing sentences or grammatical errors

RC 1) : 'a', 'an', 'the' are missing throughout the manuscript. AC 1) : I have try my best to revise grammar and usage in the resubmitted manuscript. I originally wanted to ask a professional service to solve the grammatical problems, but I am not sure whether this revision is the last version. If the Referee think that there is only a grammatical problem in final version, I will ask a professional agency to solve it again. Please understand my difficulties.

RC 2) : The sentence in lines 16-18. AC 2) : Lines 14-18, the sentence "The analysis indicated that the Shoulder is created by a dawn-dusk convection electric field intensity, sharp reduction and spatial nonuniform manifested. As, combination of the plasmaspheric rotation rate speed up with L-shell increase and plasma flux do radial outflow in the predawn sector to interact, and produce an asymmetric bulge that rotates eastward. " is replaced by "The analysis indicates that the Shoulder is created by sharp reduction and spatial nonuniform of a dawn-dusk convection electric field intensity. Combined action of the plasmaspheric rotation rate speeding up with L-shell and plasma flux doing radial outflow in the predawn sector, results in an asymmetric bulge rotating eastward to reproduce the Shoulder structure. "

RC 3) : The sentence in lines 73-74. AC 3) : Lines 73-74, the sentence "Subsequent pictures show that the Shoulder-like structure remaining and corotating with main plasmaspheric body by discussion in the next section." is replaced by " Comparison sequential observations with the simulation pictures, show that the Shoulder-like structure keeping and corotating with the main plasmaspheric body can be seen in Figure 3, and is discussed in the next section".

RC 4) : The sentence in lines 79-80. AC 4) : Lines 79-80, the sentence " In the next section, we would discuss simulation of Shoulder and Plume evolution on 8 June 2001 case base on the TPM method " is replaced by " In the next section, we take the case of 8 June 2001 observation as an example, to discuss the simulation of the Shoulder and the Plume evolution based on the TPM method. "

RC 5) : Line 105: Word->World AC 5) : Line 106, the word "Word" is replaced by " World ".

RC 6) : The sentence in lines 79-80. AC 6) : The same as RC 4).

RC 7) : Line 109: run-> runs AC 7) : Line 110, the word "run " is replaced by " runs ".

RC 8) : Line 110: which-> whose AC 8) : Line 111, the word "which " is replaced by " whose ".

RC 9) : The sentence in lines 148-150. AC 9) : Lines 148-150, the sentence "The Shoulder1 firstly arises at 12 UT in the morning sector( see in Fig.3(a)), and then corotates with the Earth reaching to the afternoon region at 18 UT ( see in Fig.3(c)), on 8 June 2001. At this time, Kp index increases to 3+ " is replaced by " The Shoulder1 firstly arises on Fig.3(a) in the morning sector ( at 12 UT, 8 June 2001 ), and then corotates with the main body of the plasmasphere to the afternoon sector on Fig.3(c)( at 18 UT, 8 June 2001 ). During this period, Kp index increases to 3+ from 1"

RC 10) : Line 156: the infantile Plume2. What does 'infantile' mean? AC 10) : 'the infantile Plume2' means the Plume2 just appear, not mature Plume structure in line

158.

RC 11) : The sentence in lines 168-169. AC 11) : Lines168-169, the sentence "The plasma refilling from plasma sheet results in the Notch structure disappear (Gallagher et al., 2005). The results of simulation show the Channel structure in Fig.3(e)-(f) " is replaced by "Plasma refilling originating from plasma sheet, result in the Notch structure disappearance (Gallagher et al., 2005). The results of simulation reproduces the Channel structure in Fig.3(f) ".

RC 12) : The sentence in lines 148-150.. AC 12) : The same as RC 9).

RC 13) : The sentence in lines 175. AC 13) : Lines173-175, the sentence " due to the fact that the potential structure does not cause the inward convection of plasma in the afternoon sector, and the low disturbance time is maintained for no long enough time. " is replaced by " due to the fact that the potential structure not cause the inward convection of plasma in the afternoon sector, and the low disturbance time is maintaining for not long enough."

RC 14) : The sentence in lines 184-187. AC 14) : Lines184-187, the sentence "The Bz value must lower than previous 24-hours value, due to the intensity of the convection electric field lower than previous level, so the last closed equipotential line (LCE) would close to the Earth and result in plasmapause of peeled off in the predawn sector (Zhang et al., 2013). One can see that no shoulder appearance in the results of the simulation, produced at 02:00 UT, 05:00 UT, and 08:00 UT on 9 June 2001 respectively. " is replaced by " One can see that no shoulders reproduced in the results of the simulation, at 02:00 UT, 05:00 UT, and 08:00 UT on 9 June 2001 respectively. The Bz value of southward component must less than previous 24-hours mean value. The intensity of the convection electric field is greater than previous 24-hours level. So the last closed equipotential line (LCE) would closer to the Earth and results in plasmapause of peeled off in the predawn sector (Zhang et al., 2013). "

RC 15) : The sentence in lines 208-210. AC 15) : Lines208-210, the sentence " So,
the Shoulder has a sharp eastern edge about 0.5Re∼0.7Re in radial extension and in a range of 3 MLT." is replaced by " So, the Shoulder has a sharp eastern edge about 0.5Re∼0.7Re in radial extension and across a narrow 3-5 hours MLT region "

RC 16) : The sentence in lines 218-220. AC 16) : Lines218-220, the sentence "The previous researchers analyze the EUV observation and propose the Shoulders structure have MLT sharpening in the angular direction, which indicate the outer edge of the Shoulder rotates faster than the inner edge, resulting in the gradual increase of MLT-profile of the Shoulder (Goldstein et al., 2002) " is replaced by " The previous researchers analyzed the EUV observation and proposed the Shoulder structure has MLT sharpening in the angular direction. It indicates that the outer edge of the Shoulder rotates faster than the inner edge, resulting in steepening of the MLT-profile of the Shoulder (Goldstein et al., 2002). "

RC 17) : The sentence in lines 239-240. AC 17) : Lines 239-240, the sentence "So, we suggest that the physical mechanism of shoulder formation is the result of plasma extrusion in the predawn sector, caused by outer plasmasphere drifts radial outward and rotates faster." is replaced by " So, we propose that the physical mechanism of the shoulder formation is plasma extrusion of outer plasmasphere in the predawn sector, due to outer plasmasphere both drifts radial outward and rotates faster. "

RC 18) : The sentence in lines 244-246. AC 18) : Lines 244-246, the sentence " The first reason is that the level of Kp index and the convection of magnetosphere is increase, so the value of these parameters driven convection field in this case is greater than the previous study articles in the geomagnetic quite case" is replaced by "The first reason is that this is a substorm case, so the convection of magnetosphere is greater than the previous study articles of the geomagnetic quiet case. "

RC19) : Line 255:downside-> dawnside AC 19) : Line 255, the word "downside " is replaced by " dawnside ".

RC 20) : The sentence in lines 218-220. AC 20) : The same as RC 16)

You can see the detailed changes in the resubmitted manuscript. If you have any problems, please contact me immediately. I am very grateful for your comment. Thank you very much.

Best Regard Hua Zhang The 1th author of this manuscript

Please also note the supplement to this comment:
https://angeo.copernicus.org/preprints/angeo-2020-86/angeo-2020-86-AC1-supplement.pdf

---

## Author Comment (AC2) · 9 Mar 2021

Dear anonymous Referee2:

I am very happy to receive your recommendation and very grateful for your advice. We make effort to revise this manuscript following your comments. Then, due to poor English make readers understand difficulty, we make efforts to revise and hope that you could be satisfied. In the resubmitted paper, new text is emphasis as red text.The Referee Comments is abbreviated to "RC", and Authors' Response is abbreviated to "AR".

The following are the response of major comments: RC 1: Line 83: Plasmasphere ions are defined in the introduction as having energies of less than 1 eV. Here the definition

given is that the plasmasphere consists of "several eV or less". The two descriptions need to agree; <1 eV is generally used, but there is flexibility. Citing a source for whatever is used is worthwhile. AR 1:' I am agree with the advice, and have revised this problem in line 86.

RC 2: Line 85: It is stated that the intensity of the electric field model is a superposition of the convection and corotation electric fields. The electric field model this line refers to is not directly stated. Line 54 states that the TPM uses the Weimer statistical electric field model. The Weimer model is empirical, based on observations. It is not a superposition of simple electric fields. The extent to which the Weimer model accurately represents measured electric fields and those measurements accurately represent actual electric fields, then this empirical model incorporates all the physical processes that produce large to small-scale features in inner magnetospheric electric fields. It also means that nothing about the underlying physical processes are available to be determined by use of the Weimer model in the TPM simulation. This point is of particular importance in the Section 4 Discussion and in the conclusions stated for the paper. AR 2: Line 85 state that the the intensity of E-model is the electric field in formula $E \times B/B2$. Where after, lines 87-89 states that the TPM uses the Weimer statistical electric field model as the convection electric field. Other convection electric field have been used to study evolution of the Plume and the Shoulder structure (like Pierrard and Cabrera, 2006; Pierrard, et al., 2008, used E5D model). So, I think that the Weimer electric field model mapping to inner magnetosphere can use in the TPM simulation.

RC 3: Line 94: Only 10 particles per simulation box is quite course. What are the boundary conditions for the simulation? Are particles allowed to leave or enter the simulation to maintain the number within the simulation? Only black and white is used to represent model results in Figure 3. If black means there is at least one particle in a simulation box, then that needs to be stated. AR 3: The calculation regions is radial range of 2-7 Re, and the TMP runs 3 days under the low activity condition to obtain the boundary conditions for the simulation. The caption of Figure 3 is rewrote.
RC 4: Line 96: Why is it stated that the density variation goes as LËĘ-3? Most authors report a variation of approximately LËĘ-4. Whatever is used here needs to be justified in the text or by citation. AR 4: I am agree with the advice, and have revised " LËĘ-3" to "LËĘ-4".

RC 5: Lines 114-116: Can this simulation produce a smooth plasmapause boundary when there are so few particles in the simulation? What is considered to be smooth given the small number of particles in each simulation box? AR 5: I rewrite this sentence " The results of EUV observation show that the plasmapause is seldom smooth or irregular, due to the fluctuations in plasmapause region cause by successive particles injection during a disturbance period (Goldstein et al., 2002; Gallagher et al., 2005), in agreement with previous whistler observations (Carpenter and Anderson,1992). Contrary, The simulation of plasmapauses by TPM is better smooth". The observation ( red lines in Figure 3) is no smooth. The simulation boundary is a skeleton ( also called an artifact ) which consists of continuous particles distribution

RC 6: Lines 114-116: What particle injections are referred to here? An "injection" of particles would normally be expected to come from outside the simulation, whether along the field or transverse to it. Quantitatively, what does smooth or irregular mean as it is used here and how can it be "seldom smooth or irregular" as stated? AR 6: An "injection" of particles reported by Goldstein et al (2002) and Gallagher et al (2005) come from plasma sheet. And "seldom smooth or irregular" is small-scale structure of the plasmapause explained in AR5.

RC 7: Line 129: How is a sharper plasmapause boundary model result shown in Figure 3? "Sharper" has previously been used to qualitatively refer to the density gradient across the plasmapause. The black and white representation of the model result shown in Figure 3 cannot show a gradient. Small irregularities in the plasmapause can be seen in Figure 3, however this may be due to the small number of particles in simulation cells, a modeling artifact rather than a physical result. AC 7: I am agree with the advice. The "Sharper"represent the observation, and has been deleted. So, the sentence
revised to "One can see that the plasmapause is closer to the Earth in the predawn sector ."

RC 8: Lines 129-131: The model result in Figure 3 does not show peeling off of plasma-spheric plasma in the predawn region. The formation of a shoulder does not constitute a peeling off of the plasmasphere. The plumes evident in all panels of Figure 3 show the plume extending sunward in afternoon and early evening magnetic local time. I am unaware of any observation of the outer plasmasphere being peeled away at predawn magnetic local times. The post-dawn outer plasmasphere has been found to some-times contribute to a broad early-plume that then narrows to afternoon MLT only. AC 8: I am agree with the advice. This is a wrong statement. "peeling off" is replaced by " inward flow". "The reason is the increase of rotation velocity resulting in plasmapause of inward flow in the predawn sector ." is rewrote in revised manuscript.

RC 9: Lines 352-354: I do not find this reference in the journal as cited. Could there be an error in the citation? AC 9: This reference is cited in lines 90-92. the sentence is "the Weimer's electric field (Weimer, 2001) is mapped into the magnetosphere along magnetic lines to model the magnetospheric convection electric field (Zhang et al., 2012)"

RC10: Lines 141-142: The plume features shown in Figure 3 exist before the shoulder convects into afternoon magnetic local times where the plume(s) is(are) connected to the plasmasphere. The shoulder is first indicated in Figure 3c near 9 hours magnetic local time. In panel (d) three hours later a plume is forming between roughly 16 hours and 18 hours magnetic local time. This shoulder feature has not azimuthally convected further than about 13 hours MLT. Another 3 hours later in panel (c) a plume appears to be forming where the shoulder has come to be located. That does not mean the shoulder had a causal role in forming the plume. It is more likely it only happened to be there when geomagnetic activity increased, which changed the global convection pattern in your electric field model so as to form a new plume that would have formed whether the shoulder was there or not. A specific explanation must be provided in

order to substantiate the statement that the shoulder is functionally responsible for the plume as currently stated. AC10: I am agree with the advice. This is only a case study. The lines 142-147 is same to your statements. So, Lines141-142 have revised to "The results of the simulation also reproduce the formation and the evolution of the Plumes,which derives from the Shoulder structure in this case, illustrated in panels of Fig.3 (d)-(f)."

RC 11: Lines 142-143: The simulation shows that the TPM simulation for the conditions during this event period resulted in a shoulder forming in at post-midnight MLT. One simulation cannot establish a pattern of shoulders forming at post-midnight MLTs as currently stated. AC11: The shoulder forming in at post-midnight MLT also have been investigated by Goldstein et al.( 2002 ), Verbanac et al. (2018), Pierrard and Cabrera (2005) , and so on. The simulation use TPM to study the formation of the Shoulder and get same results. In section 4, Trace test particles and obtain the conclusion that the shoulder arises at 3 hours MLT and explained by differential rotation rate.

RC12: Lines 148-150: The feature at 12 hours MLT in Figure 3a appears to be a remnant plume originally formed in at afternoon MLT due to earlier activity. It does not have the characteristics of a shoulder. The discussion in the last paragraph on page 8 is at least poorly expressed if not also poorly conceived, as suggested in the previous few comments. It needs to be corrected or removed. AC 12: I am agree with the advice. The shoulder adheres to the main plasmasphere, not is a deciduous remnant plume. Lines 148-153, the sentences have replaced by "The Shoulder1 firstly arises on Fig.3(a) in the morning sector ( at 12 UT, 8 June 2001 ), and then corotates with the main body of the plasmasphere to the afternoon sector on Fig.3(c)( at 18 UT, 8 June 2001 ). During this period, Kp index increases to 3+ from 1 ( see in Fig.2), and magnetosphere convection slightly enhance that triggers plasma elements in the Shoulder1 doing sunward convection, then produces the Plume1 at 21 UT on 8 June 2001 (see in Fig.3(d))"

RC 13: Line 168: Notch structures and the outer plasmasphere do not refill from the

plasma sheet as currently stated in the text. The injection of plasma ions discussed by Gallagher et al. (2005) refers to a potential source of meso-scale electric field due to charge separation in the injected energetic plasma. It is suggested in that study that this meso-scale electric field may cause the interior "W" shaped feature. AC 13: I am agree with the advice and delete the sentence "Plasma refilling originating from plasma sheet, result in the Notch structure disappearance (Gallagher et al., 2005)." in revised manuscript.

RC14: Line 173: What is meant by "inward convection?" Convection inward to lower L-shell does not appear to happen during storm-time recovery. Isopotential contours are not axially symmetric, however, therefore there can be inward and outward radial motion of the plasmapause without a change in plasmasphere content. The dusk bulge is an example. AC 14: I am agree with the advice. The"inward convection"is replaced by "inward flow".

RC 15: The first paragraph of the Discussion section: Figure 4a shows paths taken by semi-corotating plasma, but does not show the formation of a sharp radial change in the plasmapause as stated. Goldstein et al. (2002) state that their model shows the shoulder forming across a narrow 3-5 hour MLT region. Figure 4a shows a gradual outward motion of a radial group of particles from 3-9 hours MLT. For a shoulder to form there must be a transient and narrow MLT region where plasma is differentially moved in L-shell, which cannot be shown with the test particle simulation presented. Even if an adjacent and leading parcel of plasma did not move outward in L-shell after 1300UT, only a difference of 0.1-0.2L appears to have taken place between 3-5 MLT, not 0.5-0.7L as stated. Figure 3b-c show shoulder-2 formation much better, though a shorter time interval between these panels would do that much better. In fact a sequence of model images between these two might provide a more useful display than the current Figure 4a. Reversal of relative semi-corotation with L-shell shown in 4b is interesting as an explanation for steepening the leading eastward edge of a shoulder. AC 15: I am agree with the advice. The shoulder forming across a at 03:00-06:00 MLT region

(between blue vertical line and black vertical line in Figure 4(a)). The outermost particle move outward 0.7 L, and the fourth particle move outward 0.45 L, from 03:00 MLT to 08:00 MLT. the Shoulder's sharp eastern edge is differential L-shell. So, I revised it as "the Shoulder has a sharp eastern edge about 0.2Re~0.3Re in radial extension and across a narrow 3-5 hours MLT region" in the manuscript.

RC16: Line 220: "increase of the MLT-profifile of the shoulder" does not say what is needed here. Perhaps "steepening of the MLT-profifile of the shoulder" would be a better word to use. AC16: I am agree with the advice, and the sentence is replace by "It indicates that the outer edge of the Shoulder rotates faster than the inner edge, resulting in steepening of the MLT-profile of the Shoulder".

RC 17: Lines 247-260: Which model does not include electric field shielding in the inner magnetosphere? Is this referring to your TPM? If so, the statement is not substantiated in the text. While shielding is not explicitly included in the Weimer electric field model, the fact that the Weimer model is empirical means that the model includes whatever physical processes are active. That will include shielding if it is happening, as discussed in previous papers. The Weimer model, hence TPM, provide no information about the physical processes taking place that produce the measured electric field. AC 17: I have revised the sentences 247-249 according to your advice. "And the second reason is that the Weimer electric field model is larger in practice, which results in a larger total electric field value in calculation (Goldstein et al., 2002; Pierrard et al., 2008) "

RC 18: Lines 259-260: Gallagher et al. (2005) specifically report not finding a day-night asymmetry in subcorotational flow. They also do not report finding supercorotational flow, only speculate that asymmetry in the dawn-dusk convection pattern may cause net subcorotational motion. AC18: I am agree with the advice, and deletes "supercorotational flow" in line 260.

RC 19: Conclusions: I posit that you cannot investigate the physical mechanisms for

shoulder formation using the Weimer empirical electric field model, as that model only represents the measured net electric fields resulting from whatever physical processes are involved in their formation without distinction for those processes. Please substantiate how this can be done. AC1 9: The TPM uses the convection electric field which derives from the Weimer electric field mapping into the magnetosphere along magnetic lines. Other convection electric field have been used to study evolution of the Plume and the Shoulder structure (like Pierrard and Cabrera, 2006, used E5D model; Pierrard, et al., 2008, used Weimer model and Volland-Stern model). So, I think that the Weimer electric field model can use in the TPM simulation.

RC20: Conclusion 1: It has not been demonstrated that IMF Bz must be lower than the previous 24 hours for a shoulder to form. A statistical study or theory is required before there is adequate basis for the conclusion. The statement on lines 184-186 is an observation that might be used to suggest correlation or dependence, but no more than that. AC20: I am agree with the advice and delete it. I revises conclusion 1 merged into conclusion 2.

RC 21: Conclusion 2: The conclusion does not add to what has previously been found. It is incumbent on you to be clear how this new work contributes in the context of previous work. This statement does not do that. AC21: The conclusion 2 is rewrote as "The formation of Shoulder is association with IMF northward turning in the predawn sector. And the physical mechanism of Shoulder formation is the result of plasma extrusion in the predawn sector, caused by outer plasmasphere drifts radial outward and rotates faster. Reversal of corotation rate with L-shell in post-midnight sector compares with corotation rate in midnight sector. So, the shoulder forming across a at 03:00-06:00 MLT region. "

RC 22: Conclusion 3: No significance has been established between the position of a shoulder and the formation of a plume connected to the plasmasphere at that location. Given that plumes form at the onset of convection enhancement, which is not connected to the earlier formation of a shoulder, the presence of a shoulder where

a plume begins to form is likely no more than coincidence. It is well established that plumes form in the afternoon/dusk region without the presence of a shoulder feature. AC22: Conclusion 3 is rewrote as "The formation and evolution of Plume and Channel have also been reproduce in this case. One can see single or double Plumes appear in the dusk or afternoon sector, and then become thinner with time, finally disappear. "

RC 23: Line 75: "Shoulder-like structure" is acceptable, but "shoulder-like" is not used by itself and if it is shoulder-like, then it would be better to simply refer to it as a shoulder. Lines 66-68 refer to the shoulder structure and define it in words and in Figure 1. That is adequate to subsequently refer to it as a "shoulder". Unless you consider the feature you are referring to as something different from what has previously been described as a plasmaspheric shoulder, then I recommend you simply use that description or just shoulder. AC23: I am agree with the advice, and "Shoulder-like structure" is replaced by "Shoulder structure".

RC24: Lines 12-14: This sentence is not grammatically correct. Perhaps the authors intend it to be something like, "The plasmapause formation is simulated using the Test Particle Model (TPM), which is based on drift motion, which reproduces various plasmapause structures and evolution of the Shoulder feature." AC 24: I am agree with the advice, and have revised this sentence in my manuscript.

RC25: Lines 14-18: These sentences are grammatically incorrect. English language usage needs to be improved throughout in the paper. No further comment about that will be made in this review. AC25: Lines 14-18, the sentence is replaced by "The analysis indicates that the Shoulder is created by sharp reduction and spatial nonuniform of a dawn-dusk convection electric field intensity. Combined action of the plasmaspheric rotation rate speeding up with L-shell and plasma flux doing radial outflow in the predawn sector, results in an asymmetric bulge rotating eastward to reproduce the Shoulder structure. "

RC 26: "Plume" is misspelled in Figure 3e as "plumer". AC26: "Plumer" in Figure 3e is

replaced by "Plume1 and Plume2 "

You can see the detailed changes in the resubmitted manuscript. If you have any problems, please contact me immediately. I am very grateful for your comment. Thank you very much.

Best Regard Hua Zhang The 1th author of this manuscript
* * *

---

## Referee Report (RR1)

Second Review of "A New Perspective and Explanation to the Formation of Plasmaspheric Shoulder Structure" by Zhang et al

The paper provides an interesting contribution to the discussion of shoulder formation.

Suggested language edits for the abstract are provided below. I appreciate the challenge of expressing thoughts in a non-native language. Significant editing is required for language usage throughout the text. I do not see the need to capitalize the first letter of Structure or Plume anywhere in the text other than in the first word of a sentence or in a title. I highlight text using the color red to indicate suggested language edits in the title and abstract.

- Line 1, the title, "A New Perspective and Explanation for the Formation of Plasmaspheric Density Structures"
- Line 11," and a post-noon plume-like structure .
- Line 15, "reduction and spatial non-uniformity in the dawn-dusk convection electric field intensity."
- Lines 16-18, "A TPM modeled event is found to develop an initial pre-dawn asymmetric bulge that becomes a shoulder as a result of increased "co-rotation" rate with increasing L-shell that is preceded by localized outward convection."
- Lines 18-20, "The shoulder rotates eastward toward 12h MLT and develops into a single or double plume structure during an active time period.

---

## Author Response (AR2)

Dear anonymous Referee1:

I am very happy to receive your recommendation and very grateful for your advice. We have followed your comments to revise this manuscript. The grammar and language usage of the manuscript have been revised by a native-English speaker. In the resubmitted paper, a new text is emphasis as red text. The Referee Comments are abbreviated to "RC", and Authors' Response is abbreviated to "AR".

The following are the responses of each major comment:

RC1 :

AR (1) 'I agree with the advice and have revised this problem in my manuscript. We will call for a professional company to polish the manuscript before formal publication.'

AC (1) 'Please understand my difficulties.'I understand the difficulties and efforts needed to correct the manuscript. However, it should be noted that the formal rating of the manuscript includes 'Is the language fluent and precise?' It is the reviewer's responsibility to ensure that the language meets the requirement of the journal.

Before reaching out to the 'professional company,' it is highly recommended that the authors carefully proofread the manuscript. Besides, there are free grammatical softwares (e.g., Grammarly).

AR1 :

I agree with your advice. The grammar and language usage of manuscript has been revised by a native-English speaker. In the resubmitted paper, a new text is emphasis as red text. I hope that you can be satisfied.

RC2 :

AR (2). The paper's main conclusion is that the convection electric field changed the trajectory and rotation rate, which resulted in a Shoulder structure. The TPM model is used to support this conclusion. Therefore, the following observations/data are essential to prove this conclusion: 1. The electric field should be shown. 2. The plasmasphere from the TPM model. 3. A comparison between the TPM model and real observations.

The authors kept the origin Figure 3 and added supplementary figures of the raw IMAGE/EUV. However, items 1 and 3 are still missing, making the manuscript unconvincing.

It is necessary to show the electric field's evolution, the plasmapause from the TPM model, and the real IMAGE/EUV observation. The authors can refer to the figures in the following references.

Goldstein, J., Wolf, R. A., Sandel, B. R., & Reiff, P. H. (2004). Electric fields deduced from plasmapause motion in IMAGE EUV images. Geophysical Research Letters, 31(1), L01801. doi:10.1029/2003GL018797.

Goldstein, J., Pascuale, S., & Kurth, W. S. (2019). Epoch-Based Model for Stormtime Plasmapause Location. Journal of Geophysical Research: Space Physics, 124(6), 4462-4491. doi:https://doi.org/10.1029/2018JA025996.

AR2 :

I agree with your advice. I add Figure 4 to illustrate origin observations by EUV/IMAGE and equipotential lines in the equatorial plane to supplement figure 3 in lines 143-145.

RC3 :

AR (3) Since the feature of a double plume is neither demonstrated nor observed in this manuscript, I recommend it not emphasized in the abstract.

AR3:

I agree with your advice and revised the abstract.

RC4:

Figure 4 and RC 6. AC7.

Figure 4 is still confusing to me. What do the colors mean in Figure 4b? Do they represent the test particles at different L shells (as answered in AC7)? The colors in Figure 4b should be corrected because they are not the same as the color bar in Figure 4a.

Figure 4a shows the trajectory of 14 test particles, which have different rotation rates according to Figure 4b. On the other hand, if the x-axis is both time-dependent (UT) and MLT-dependent, does it mean that these particles are co-rotating with the Earth (thus a fixed rotating rate).

AR4:

I agree with your advice and revised Figure 5, the x-axis is MLT-dependent. The time is used to drive the model running. The colors mean in Figure 4 represent the test particles at different L shells.

You can see the detailed changes in the resubmitted manuscript. If you have any problems, please contact me immediately. I am very grateful for your comment. Thank you very much.

Best Regard
Hua Zhang
The 1th author of this manuscript

Dear anonymous Referee2:

I am very happy to receive your recommendation and very grateful for your advice. We have followed your comments to revise this manuscript. The grammar and language usage of the manuscript have been revised by a native-English speaker. In the resubmitted paper, new text is emphasis as red text.The Referee Comments is abbreviated to "RC", and Authors' Response is abbreviated to "AR".

The following are the response of each major comment:

RC :

Suggested language edits for the abstract are provided below. I appreciate the challenge of expressing thoughts in a non-native language. Significant editing is required for language usage throughout the text. I do not see the need to capitalize the first letter of Structure or Plume anywhere in the text other than in the first word of a sentence or in a title. I highlight text using the color red to indicate suggested language edits in the title and abstract.

Line 1, the title, "A New Perspective and Explanation for the Formation of Plasmaspheric Density Structures"

Line 11," and a post-noon plume-like structure straddling in the between noon and dusk region.

Line 15, "reduction and spatial non-uniformity in the dawn-dusk convection electric field intensity."

Lines 16-18, "A TPM modeled event is found to develop an initial pre-dawn asymmetric bulge that becomes a shoulder as a result of increased "co-rotation" rate with increasing L-shell that is preceded by localized outward convection."

Lines 18-20, "The shoulder rotates eastward toward 12h MLT and develops into a single or double plume structure during an active times period.

AR :

I am agree with the advice to revise an abstract of my manuscript. The grammar and language usage of manuscript has been revised by a native-English speaker. The "Plume "and "Shoulder" also are replaced by "plume" and "shoulder" throughout the text.

 You can see the detailed changes in the resubmitted manuscript. If you have any problems, please contact me immediately. I am very grateful for your comment. Thank you very much.

Best Regard
Hua Zhang
The 1th author of this manuscript

---

## Author Response (AR3)

Dear editor:

I am very happy to receive your recommendation and very grateful for your advice. We have followed your comments to revise grammar and language usage of this manuscript. In the resubmitted paper, a new text is emphasis as red text.

The following are the detailed modifies:

1.  Line 28: the word "moves " is added.
2.  Line 31: the word "study" is replaced by " studies".
3.  Line 33: the word "did" is added.
4.  Line 35-38: the sentence "The EUV instrument onboard IMAGE satellite was launched in March 2000, which provided a global perspective to the plasmasphere, such as plume, finger, notch and shoulder, and so on, some of plasmaspheric structures observed by EUV (Sandel et al., 2001)." is replaced by "The EUV instrument onboard IMAGE satellite was launched in March 2000, that provided a global perspective of the plasmasphere. Such as plume, finger, notch and shoulder, and so on, were observed by EUV (Sandel et al., 2001)."
5.  Line 38: the words "has less study " is replaced by " has been less studied".
6.  Line 39: the word "But" is replaced by "However".
7.  Line 45: the word "a" is added.
8.  Line 47: the word "trigger" is replaced by "it triggers".
9.  Line 67: the word " The" is deleted.
10.  Line 69: the number "15.05" is replaced by "15:05".
11.  Line 70: the word "Figure1" is replaced by "Figure 1"
12.  Line 74: the word "of " is added.
13.  Line 75: the words "keeping and " are deleted.
14.  Line 94: the word "regions" is replaced by "region".
15.  Line 112: the word "over" is replaced by "from".
16.  Line 116: the words "simulation plasmapauses " is replaced by " simulated plasmapause"
17.  Line 117: the word "particles" is replaced by "particle".
18.  Line 118: the words "indicates that" is added, and the punctuation "," is deleted.
19.  Line 124: the word "Contrary " is replaced by "In contrast" ,and "better " is replaced by "more ".
20.  Line 137: the word "origin" is replaced by "original".
21.  Line 142: the sentence "8 June at 12:00 UT to 9 June at 09:00 UT 2001, and every three hours output a snapshot" is replaced by "at 12:00 UT on 8 June to at 09:00 UT on 9 June in 2001 with snapshots every three hours ".
22.  Line 143: the word "origin" is replaced by "original".
23.  Line 145: the word "that" is deleted.
24.  Line 164: the word "convection" is replaced by "convect".
25.  Line 169: the words "slightly enhances" is replaced by "is slightly enhanced".
26.  Line 178: the word "is " is added.
27.  Line 183: the word "And" is replaced by "Moreover".
28.  Line 186: the word "become" is replaced by "becomes", and the punctuation "," is added.

29. Line 190: the words "event, due to" is replaced by "event is due to".

30. Line 191: the word "dose" is replaced by "does".

31. Line 195: the word "Sudden" is replaced by "suddenly".

32. Line 200: the word "that" is added, and the second punctuation "," is deleted.

33. Line 221: the sentence "in the before 02:00 MLT sector" is replaced by "before 02 MLT".

34. Line 223: the sentence "The shoulder forming across a at 03:00-06:00 MLT region" is replaced by "The shoulder is forming across 03-06 MLT".

35. Line 229-230: the sentence "The conclusions of Goldstein (2002) and Verbanac (2018) verify the simulation of this paper" is replaced by "The simulation of this paper verifies the conclusions of Goldstein (2002) and Verbanac (2018)".

36. Line 262: the word "And" is deleted, and the word "the" is replaced by "The".

37. Line 282: the word "that" is added.

38. Line 289: the words "the fact that" are added, and the word "radial" is replaced by "radially".

39. Line 290: the sentence "Reversal of corotation rate with L-shell in post-midnight sector compares with corotation rate in midnight sector." is replaced by "The corotation rate in midnight sector decreases with the increasing L-shell, while it increases in pre-dawn sector."

40. Line 297-298: the sentence "expect to obtain more perfect results compared with EUV observations" is replaced by "and we expect to obtain more reasonable results".

41. We add a new author for this paper, and add the sentence "Wu Yewen gives some suggestions and draws Figure 4 for the paper." in line 303-304.

You can see the detailed changes in the resubmitted manuscript. If you have any problems, please contact me immediately. I am very grateful for your help. Thank you very much.

Best Regard
Hua Zhang
The 1th author of this manuscript

Dear anonymous Referee1:

I am very happy to receive your recommendation and very grateful for your advice. We have

followed your comments to revise this manuscript. The grammar and language usage of the manuscript is revised again. In the resubmitted paper, new text is emphasis as red text. The Referee Comments is abbreviated to "RC", and Authors' Response is abbreviated to "AR".

The following are the response of comment:

RC :

Minor point: Figure 5 (original Figure 4). The authors agreed that the x-axis is MLT, which makes sense to me. However, the text needs to be changed. E.g., line 213-215: '…. corresponding to both given MLT and UT illustrated in the bottom of Fig.5.'

AR :

Line 213-215: the sentence "Outputs are the trajectory (see in Fig.5(a)) and the rotation rate (see in Fig.5(b)) of these test particles corresponding to both given magnetic local time and universal time illustrated in the bottom of Fig.5." is replaced by "Outputs are the trajectory (see in Fig.5(a)) and the rotation rate (see in Fig.5(b)) of these test particles corresponding to given magnetic local time illustrated in the bottom of Fig.5."

You can see the detailed changes in the resubmitted manuscript. If you have any problems, please contact me immediately. I am very grateful for your comment. Thank you very much.

Best Regard

Hua Zhang

The 1th author of this manuscript